# Prevalence of double heterozygotes of HbE and α-thal 1 (SEA) type in pregnancies and their partners that received antenatal care at Chiangrai Prachanukroh Hospital and reevaluated the cut-offs for differentiation

**Prapapun Leckngam** [1,2]*, **Phanida Wamontree**[3], **Tapanut Chuaykarn**[4]

**1** School of Health Science, Mae Fah Luang University, Chiangrai, Thailand, **2** Biomedical Technology Research Group for Vulnerable Populations, Mae Fah Luang University, Chiangrai, Thailand, **3** School of Integrative Medicine, Mae Fah Luang University, Chiangrai, Thailand, **4** Antenatal Care Clinic, Chiangrai Prachanukroh Hospital, Chiangrai Thailand

* prapapun.lec@mfu.co.th

## Abstract

### Background

In Thailand, double heterozygotes of HbE and α-thal 1 (SEA) type is common. If pregnant women carrying this genotype conceive with partners who also carry the same genotype or who are carriers of α-thalassemia 1 (SEA) type, there is 25% chance of a fetus having Hb Bart's hydrops fetalis. This research aims to determine the prevalence of double SEA/E, compare the hematological parameters and HbE levels between pure HbE heterozygote and double SEA/E and reevaluate the cut-offs used to screen individuals with these genotypes.

### Methods

In a retrospective study, 397 samples from individuals (partners who visited the Antenatal Care Clinic, Chiangrai Prachanukroh Hospital, Chiangrai Province, from 2017–2021) with HbE (hemoglobin typing EA) were collected. The samples were then classified as HbE with or without α-thalassemia 1 (SEA) type and RBC, Hb, Hct, MCV, MCH, MCHC, RDW and HbE level were compared between pure HbE heterozygote and double SEA/E via Student's t-test, with a p-value of 0.05. Cut-offs were reevaluated in samples that avoided interfering with iron-deficiency anemia via the parameters of sensitivity, specificity, PPV, NPV and accuracy.

### Results

The study revealed that the prevalence of double heterozygotes of HbE and α-thal 1 (SEA) type was 12.6%. HbE level, MCV, MCH and MCHC in double heterozygotes of HbE and α-thal 1 (SEA) type were significantly lower, whereas RBC and RDW

**Data availability statement:** All relevant data are within the paper and its Supporting Information files.

**Funding:** Mae Fah Luang University Grant no. 651A05003 Mae Fah Luang University, the funder, had no role in the study design, data collection and analysis, decision to publish or preparation of the manuscript.

**Competing interests:** The authors have declared that no competing interests exist.

were greater (p < 0.001). Reevaluation for cut-offs revealed that, in the group with Hb < 10.0 g/dL, combined cut-offs (MCH + HbE level) and (MCV + MCH + HbE level) for the group with Hb = 10.0–11.9 g/dL were most effective, while a single cut-off (HbE level) was most effective for the group with Hb ≥ 12.0 g/dL.

## Conclusions

Double SEA/E were also common among pregnant women and their partners who visited Chiangrai Prachanukroh Hospital, Chiangrai, Thailand. Using appropriate cut-offs for hematological parameters and HbE levels allows pure HbE heterozygote to be distinguished from double SEA/E.

## Introduction

Thalassemias and hemoglobinopathies are important health issues in Thailand. Currently, the incidence of thalassemia syndrome in Thailand is approximately 1% (affecting around 620,000 people). On average, approximately 30–40% or 18–24 million people, are carriers. Each year, approximately 5,000 pregnant women miscarry or give birth to children with severe thalassemia including homozygous α-thalassemia 1 (Hb Bart's hydrops fetalis), homozygous β-thalassemia and HbE/β-thalassemia [1–2]. Among the β-hemoglobinopathies in Thailand, HbE is the most significant due to its high prevalence in all regions [3–4]. G-to-A substitution at codon 26 of the β-globin chain leads to HbE. This mutation converts the glutamic acid triplet codon to a lysine and triggers a cryptic splice site at codon 25, which decreases synthesis of $\beta^E$ - globin chain and causes $\beta^+$-thalassemia phenotype [5–6]. The prevalence of HbE in Thailand is high and varies by region, ranging from 14 to 60%. The frequency of HbE is 14.9–37.3% in North Thailand [7–9], 39.1-(50–60)% in Northeast [10], 22.7% in Central [2,11,12] and 16% in South [11,13]. Beside HbE, the most common α-thalassemia 1 (Southeast Asian (SEA) type), a severe form of α-thalassemia commonly found in Southeast Asian countries, including Thailand. It is caused by the deletion of an approximately 20 kb of α-globin gene cluster on chromosome 16 p13.2 [3,14]. α-thalassemia 1 (SEA) type carriers account for 5.2–21.0% in North Thailand [7–9,15], 3.1–13.6% in Northeast [2,10], 16% in Central [2] and 4.6% in South [16–17]. HbE alone does not cause significant clinical complications, but coinheritance with different forms of α-thalassemia can cause a wild range of symptoms [5–6]. The coinheritance (double heterozygotes of HbE and α-thal 1 (SEA) type) is common in Thailand and the probability of inheritance depends on an individual's ethnic background. In a study by Tatu T et al. (2012) [18], the percentage of double heterozygotes of HbE and α-thal 1 (SEA) type among HbE carriers was approximately 11.2% and Charoenkwan et al. (2005) [19] reported a value at 11.0% among pregnant women and/or their partners who visited Maharaj Nakorn Chiangmai Hospital. Limveeraprajak E (2019) studied couples considered at risk who attended Sawanpracharuk Hospital, Nakorn Sawan Province, between 2012 and 2017 and found that among carriers of HbE and/or β-thalassemia carriers, 6.3% were double

heterozygotes of HbE and α-thal 1 (SEA) type [20]. If women who carry double heterozygotes of HbE and α-thal 1 (SEA) type conceive with partners who carry the same genotype or who are carriers of α-thalassemia 1 (SEA) type, there is a 25% chance of with the fetus having Hb Bart's hydrops fetalis (Fig 1) [21]. Most affected fetuses died before, during or shortly after delivery. Preeclampsia, polyhydramnios or oligohydramnios, antepartum hemorrhage and preterm delivery are at-risks for mothers [13]. Due to the poor survival rate of an affected fetuses and the potential maternal complications, termination is usually advised. Therefore, in areas where SEA-α thalassemia 1 and HbE are prevalent, the ability to identify double heterozygotes of HbE and α-thal 1 (SEA) type is important. HbE heterozygote exhibit minimal alterations in their hematological parameters, such as mild microcytosis. However, hematological parameters such as MCV (mean corpuscular volume), MCH (mean corpuscular hemoglobin) and MCHC (mean corpuscular hemoglobin concentration) and HbE levels (hemoglobin E levels) are changed in double heterozygotes of HbE and α-thal 1 (SEA) type. For example, HbE levels measured using high-performance liquid chromatography (HPLC) were approximately 26.5 ± 1.5% in pure HbE heterozygote but 19–21% in double heterozygotes of HbE and α-thal 1 (SEA) type [6]. Investigation of double heterozygotes of HbE and α-thal 1 (SEA) type among HbE carriers is essential in at-risk partners, important for genetic counselling and can help prevent the occurrence of Hb Bart's hydrops fetalis. Additionally, it contributes to alleviating socio-economic problems for families and lowers the country's health care costs.

Recently, attempts have been made to distinguish pure HbE heterozygote from double heterozygotes of HbE and α-thal 1 (SEA) type by screening with MCV, MCH and HbE levels. Sanchaisuriya et al. (2008) reported that the combined cut-offs for distinction were MCV < 74 fL, MCH < 24 pg and HbE level <26% [22]. Charoenkwan et al. (2005) proposed an optimal cut-off for HbE level of less than 25% [19]. Leckngam et al. (2017) suggested hemoglobin concentration cut-offs: Hb < 10 g/dL, 10.0–11.9 g/dL and Hb ≥ 12.0 g/dL; this study demonstrated that the total Hb concentration affected the HbE level, MCV and MCH cut-offs [23]. To date, double heterozygotes of HbE and α-thal 1 (SEA) type, particularly among married partners, have not been reported in Chiangrai province. Therefore, the aims of this research were to retrospectively evaluate the prevalence of coinheritance of HbE heterozygote and α-thal 1 (SEA) type, to compare hematological parameters and HbE levels between pure HbE heterozygote and double heterozygotes of HbE and α-thal 1 (SEA) type and to reevaluate cut-offs classified according to hemoglobin levels in samples that avoided interfering with iron

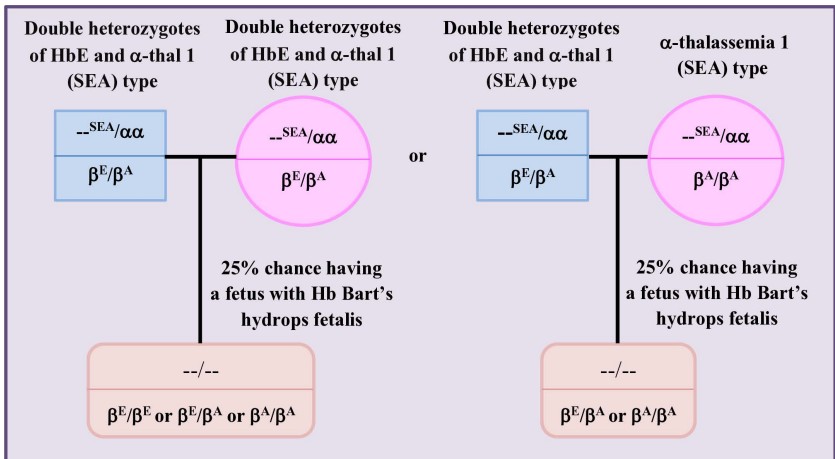

**Fig 1. Family pedigree showing the occurrence of a fetus with Hb Bart's hydrops fetalis that was found in married partners who both carry double heterozygotes of HbE and α-thal 1 (SEA) type or one carries double heterozygotes of HbE and α-thal 1 (SEA) type and another is α-thalassemia 1 (SEA) type carrier, modified from [21].**

deficiency anemia (the novelty of this study) that helps to reduce the occurrence of false positive screening (Polprasert et al., 2020) [24].

## Materials and methods

### Methods

This research is a retrospective study, based on medical records. Data were taken from pregnant women and their partners who received antenatal care at the Antenatal Care Clinic, Chiangrai Prachanukroh Hospital, Chiangrai province, from January 2017 to December 2021. This hospital serves a high volume of pregnant women which accounts for approximately 15% of all pregnancies in Chiangrai province. All individuals resided in Chiangrai or the nearby province: Phayao (Fig 2) [25]. The study received ethical approval from the Human Research Ethics Committee of Chiangrai Prachanukroh Hospital on February 20, 2023 (EC CRH 006/66 Ex) and has been exempted from the informed consent.

After permission, the data were collected from existing medical records in the Antenatal Care Clinic using a data collection form and anonymously. The collection started February 21, 2023 and ended February 10, 2024. The inclusion criteria for this study were as follows: 1) aged 18 years or older and 2) hemoglobin type EA (HbE with or without α-thalassemia) with RDW ≤ 14.5% [24]. The exclusion criteria were as follows: 1) aged less than 18 years and 2) hemoglobin type EA with an RDW > 14.5% (to exclude individuals with iron deficiency anemia) [24]. Pregnant women and their partners were screened to identify thalassemia carriers (at-risk partners) and to detect severe α-thalassemia 1 (SEA type); screening followed the guidelines of the national policy for the prevention and control of severe thalassemia in Thailand [26]. There were 27,027 pregnant women who underwent the prevention and control program for severe thalassemia. Among them, 8,154 were found to be positive for thalassemia screening. While among the partners of these women, 7,418 were found to be positive. From these partners, 397 individuals met the specific criteria: 196 males and 201 females aged 18–45 years (all data are in S1 Table).

### Screening and confirmatory tests for the detection of double heterozygotes of HbE and α-thal 1 (SEA) type

All 397 EDTA blood samples were screened utilizing a combination of the mean corpuscular volume (MCV) obtained from the complete blood count (CBC) and the results of dichlorophenol-indophenol precipitation (DCIP) test, which are widely used in Thailand [27]. DCIP can oxidize and precipitate unstable hemoglobin such as HbE because the mutation in HbE at codon 26 of the β-globin chain changes glutamic acid to lysine, which leads to a free-sulfhydryl group that can be precipitated by DCIP. Then the precipitation of HbE was evaluated with a cloudy solution [28–29]. Secondary, samples that tested positive in the screening (such as having MCV < 80 fL cut-off) and/or the positive for DCIP test, were further investigated by confirmatory tests, Capillary Electrophoresis (CE) and/or DNA analysis using Real-Time PCR with melting curve analysis (for identification of α-thalassemia 1 SEA type) (S2 Methods). Quality control in the laboratory for CE and Real-Time PCR tests is by using internal controls and participating in the Proficiency Testing Program.

### Statistically analysis

All data were analyzed using the standard statistical software package, statistics for Windows, SPSS version 25. Statistical comparison of red blood cell count (RBC count), hemoglobin concentration (Hb), hematocrit (Hct), mean corpuscular volume (MCV), mean corpuscular hemoglobin (MCH), mean corpuscular hemoglobin concentration (MCHC), red cell distribution width (RDW) and hemoglobin E level (HbE level) between pure HbE heterozygote and double heterozygotes of HbE and α-thal 1 (SEA) type was performed via Student's t-test with a p-value of 0.05 (because the parameters that were compared, included in diagnostic tests). Parameters found to be with $p < 0.05$ were considered to be statistically significant differences.

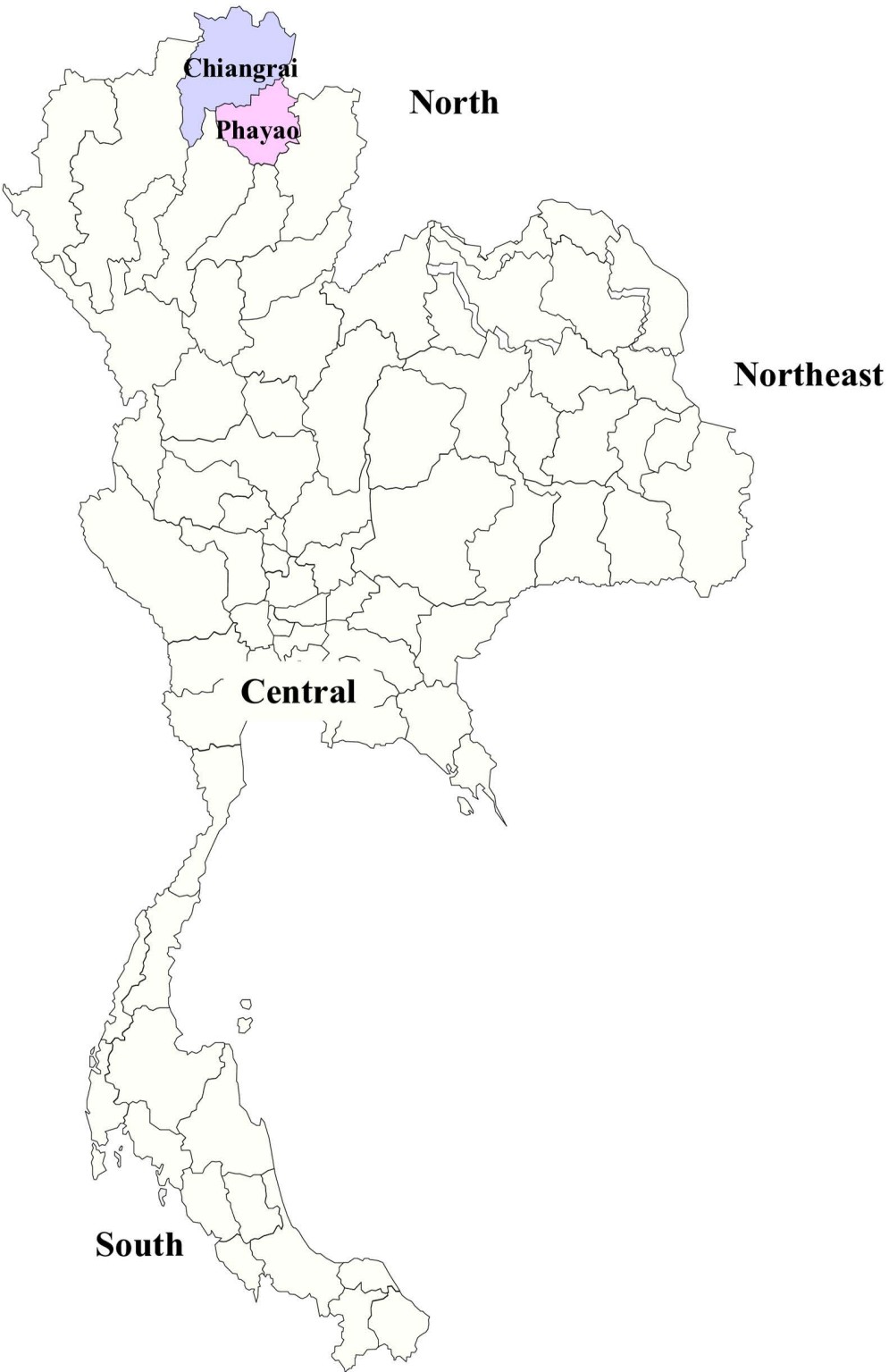

**Fig 2. Thailand map showing the provinces where the prevalence of double heterozygotes of HbE and α-thal 1 (SEA) type was studied [25].**

**Reevaluation of the cut-offs reported by Leckngam et al. (2017), for the differentiation of double heterozygotes of HbE and α-thal 1 (SEA) type from pure HbE heterozygote**

Single and combined cut-offs classified according to hemoglobin levels (Receiver Operation Curve (ROC) analysis was performed to determine the optimal cutoff values) as reported by Leckngam et al. (2017) (Hb < 10.0 g/dL, 10.0–11.9 g/dL and > 12.0 g/dL) were tested in 397 HbE heterozygotes with or without α-thalassemia. The cut-offs used to differentiate double heterozygotes of HbE and α-thal 1 (SEA) type from pure HbE heterozygote. The parameters included sensitivity, specificity, positive predictive value (PPV), negative predictive value (NPV) and accuracy.

## Results

**Comparison of hematological parameters and HbE levels in pure HbE heterozygote and double heterozygotes of HbE and α-thal 1 (SEA) type**

The data from pregnant women and their partners who visited the Antenatal Care Clinic at Chiangrai Prachanukroh Hospital and those that had hemoglobin type EA (HbE with or without α-thalassemia) were collected. These included samples from 397 individuals: 196 males and 201 females, aged 18–45 years (Table 1, S1 Table). The 397 samples were grouped according to HbE heterozygosity with or without α-thalassemia 1 (SEA) type. The prevalence of double heterozygotes of HbE and α-thal 1 (SEA) type was 12.6% and the gene frequency is shown in Table 1. The mean ± SD values of the MCV, MCH, MCHC and HbE level were lower in double heterozygotes of HbE and α-thal 1 (SEA) type than in pure HbE heterozygote (p < 0.001). In pure HbE heterozygote, MCV = 78.2 ± 3.9 fL, MCH = 24.6 ± 1.5 pg, MCHC = 27.9 ± 2.8 g/dL and HbE level = 28.1 ± 2.8%, while in double heterozygotes of HbE and α-thal 1 (SEA) type, MCV = 65.5 ± 7.1 fL, MCH = 22.4 ± 2.9 pg, MCHC = 25.3 ± 2.7 g/dL and HbE level = 18.9 ± 3.4% (Fig 3, Table 2). However, the RBC count and RDW were higher in the double heterozygotes of HbE and α-thal 1 (SEA) type (5.33 ± 0.49 million/uL and 13.7 ± 0.4%, respectively) than in pure HbE heterozygote (4.53 ± 0.30 million/uL and 13.4 ± 0.6%, respectively) (p < 0.001; Fig 3, Table 2, S3 Table). The Hb and Hct did not statistically differ (p > 0.05) within those in the double heterozygotes of HbE and α-thal 1 (SEA) type (Hb = 11.6 ± 0.8 g/dL, Hct = 34.9 ± 2.5%) being slightly lower than those in pure HbE heterozygote (Hb = 11.8 ± 0.7 g/dL, Hct = 35.3 ± 2.2%) (Fig 3, Table 2, S3 Table). Comparison stratified by gender as shown in Table 3 (S4 Table).

**Reevaluation of the cut-offs used to differentiate pure HbE heterozygote from double heterozygotes of HbE and α-thal 1 (SEA) type**

Single and combined cut-offs classified according to hemoglobin levels as reported by Leckngam et al. (2017) (Hb < 10.0 g/dL, 10.0–11.9 g/dL and > 12.0 g/dL) were used to differentiate double heterozygotes of HbE and α-thal 1 (SEA) type from pure HbE heterozygote. There were 10 individuals in the group with Hb < 10.0 g/dL, 230 individuals in the group with Hb = 10.0–11.9 g/dL and 157 individuals in the group with Hb ≥ 12.0 g/dL. The sensitivity, specificity, PPV, NPV and accuracy were 100, 100, 100, 100 and 100% in the group with Hb < 10.0 g/dL, when using the MCH + HbE level cut-offs; 100, 67.7, 30.9, 100 and 71.7% in the groups with Hb = 10.0–11.9 g/dL when using MCV + MCH + HbE level cut-offs; and

**Table 1. Age range, number, prevalence and gene frequency of pure HbE heterozygote and double heterozygotes of HbE and α-thal 1 (SEA) type among pregnant women and their partners who visited Chiangrai Prachanukroh Hospital from 2017 to 2021 (S1 Table).**

| Age range (year) | Person | Number (n = 397) | Pure HbE heterozygote (n = 347) | Double heterozygotes of HbE and α-thal 1 (SEA) type (n = 50) | Gene frequency | |
|---|---|---|---|---|---|---|
| | | | | | E | SEA |
| 18-43 | Pregnant women | 201 | 175 | 26 | 0.470 | 0.059 |
| 19-45 | Male partners | 196 | 172 | 24 | | |
| Total | | 397 | 347 | 50 | | |

**Note; E: HbE gene, SEA: α-thal 1 (SEA) type gene**

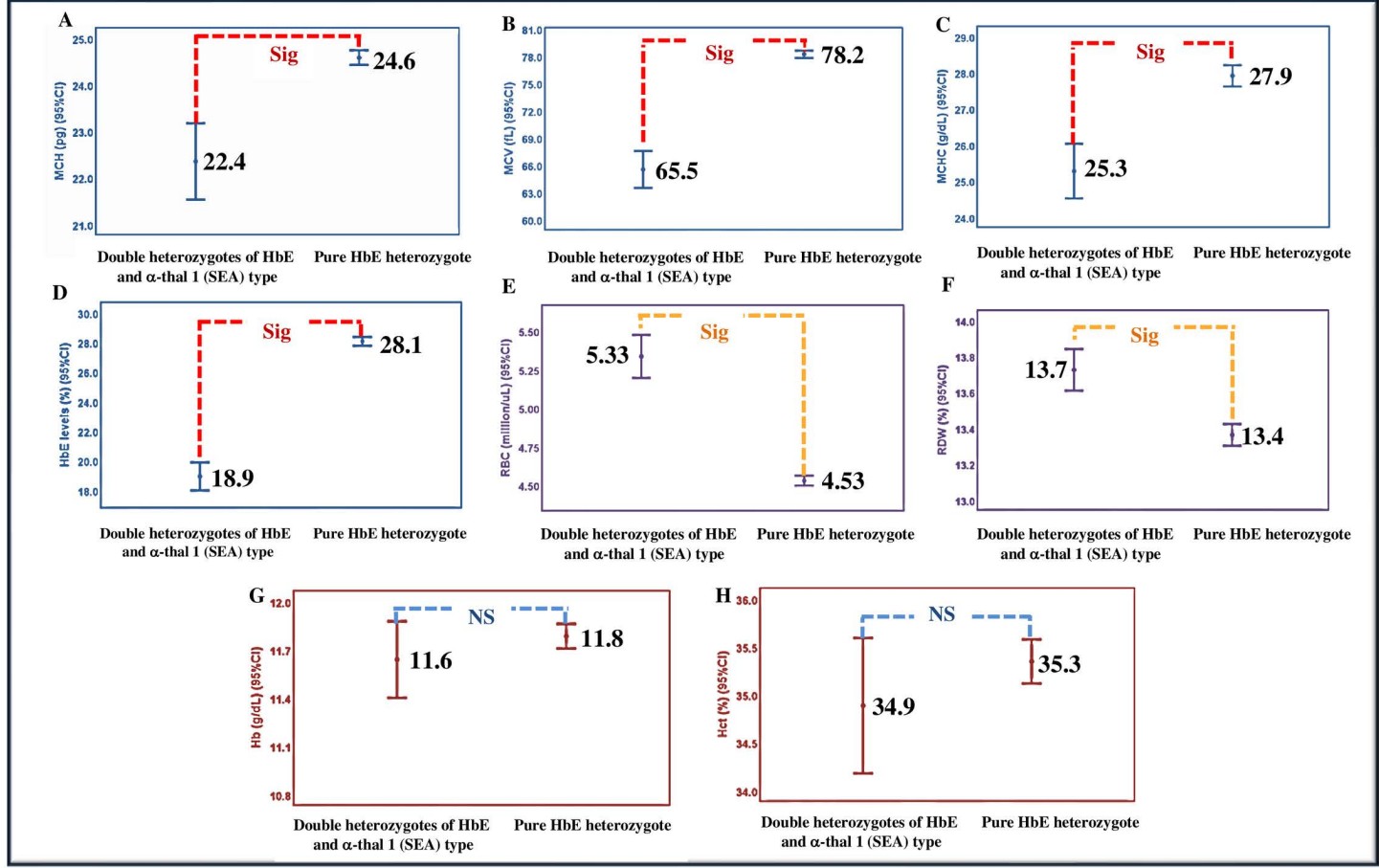

**Fig 3. Comparison of MCH (A), MCV (B), MCHC (C), HbE levels (D), RBC (E), RDW (F), Hb (G) and Hct (H) between double heterozygotes of HbE and α-thal 1 (SEA) type and pure HbE heterozygote.** The mean values are placed in the middle of each bar. "Sig" indicates a significant difference at p < 0.001 and "NS" indicates a nonsignificant difference at p > 0.05 (S3 Table).

100, 31.2, 16.7, 100 and 39.5% in the group with Hb ≥ 12.0 g/dL when using MCV + HbE level (Table 4, S5 Table). When reevaluation was performed using a single cut-off for MCH, the sensitivity, specificity, PPV, NPV and accuracy were 100, 100, 100, 100 and 100%, respectively, in the group with Hb < 10 g/dL (similar to the results obtained when using MCV and HbE level cut-offs). In the group with Hb 10.0–11.9 g/dL, the sensitivity, specificity, PPV, NPV and accuracy were 75.9, 74.1, 29.7, 95.5 and 77.1%, when using a single cut-off for MCH; 86.2, 92.0, 61.0, 97.9 and 91.2% for MCV; and 93.1, 92.0, 62.8, 98.3 and 92.2% for HbE level (Table 5, S5 Table). In the group with Hb ≥ 12.0 g/dL, the sensitivity, specificity, PPV, NPV and accuracy were 68.4%, 33.3%, 12.4%, 88.5% and 37.6% when using a single cut-off for MCH; 94.7%, 63.0%, 26.1%, 98.9% and 66.9% for MCV; and 100, 61.6, 26.4, 100% and 66.2% when using a single cut-off for HbE level (Table 5, S5 Table).

## Discussion

The coinheritance of HbE with different forms of α-thalassemias causes varying degrees of disease severity and changes in hematological parameters, including HbE levels. In Thailand, the coinheritance of HbE and α-thalassemia 1 (SEA) type is common, with a prevalence of 11.2% reported by Tatu T et al. (2012) [18] and 11.0% reported by Charoenkhuan

**Table 2. Comparison of hematological parameters and HbE levels between pure HbE heterozygote and double heterozygotes of HbE and α-thal 1 (SEA) type (S3 Table).**

| RBC parameters | Pure HbE heterozygote (n = 347) | Double heterozygotes of HbE and α-thal 1 (SEA) type (n = 50) | Student's t-test p-value |
|---|---|---|---|
| RBC count (million/uL) | 4.53 ± 0.30 | 5.33 ± 0.49 | <0.001 |
| Hb (g/dL) | 11.8 ± 0.7 | 11.6 ± 0.8 | 0.193 |
| Hct (%) | 35.3 ± 2.2 | 34.9 ± 2.5 | 0.170 |
| MCV (fL) | 78.2 ± 3.9 | 65.5 ± 7.1 | <0.001 |
| MCH (pg) | 24.6 ± 1.5 | 22.4 ± 2.9 | <0.001 |
| MCHC (g/dL) | 27.9 ± 2.8 | 25.3 ± 2.7 | <0.001 |
| HbE levels (%) | 28.1 ± 2.8 | 18.9 ± 3.4 | <0.001 |
| RDW (%) | 13.4 ± 0.6 | 13.7 ± 0.4 | <0.001 |

Note: RBC count: red blood cell count, Hb: hemoglobin level, Hct: hematocrit, MCV: mean corpuscular volume, MCH: mean corpuscular hemoglobin, MCHC: mean corpuscular hemoglobin concentration, HbE levels: hemoglobin E levels and RDW: red cell distribution width.

**Table 3. Comparison of hematological parameters and HbE levels between pure HbE heterozygote and double heterozygotes of HbE and α-thal 1 (SEA) type, stratified by gender (S4 Table).**

| RBC parameters | Pure HbE heterozygote (n = 347) | | Double heterozygotes of HbE and α-thal 1 (SEA) type (n = 50) | |
|---|---|---|---|---|
| | Male (n = 172) | Female (n = 175) | Male (n = 24) | Female (n = 26) |
| RBC count (million/uL) | 4.56 ± 0.33 | 4.50 ± 0.28 | 5.30 ± 0.52 | 5.36 ± 0.47 |
| Hb (g/dL) | 11.9 ± 0.8 | 11.7 ± 0.7 | 11.6 ± 1.1 | 11.7 ± 0.6 |
| Hct (%) | 35.7 ± 2.3 | 35.0 ± 2.0 | 34.8 ± 3.2 | 35.0 ± 1.6 |
| MCV (fL) | 78.4 ± 3.8 | 77.9 ± 4.0 | 65.3 ± 8.7 | 65.7 ± 5.6 |
| MCH (pg) | 24.4 ± 1.4 | 24.7 ± 1.5 | 21.9 ± 3.7 | 22.8 ± 1.8 |
| MCHC (g/dL) | 27.9 ± 3.1 | 27.9 ± 2.4 | 25.2 ± 3.4 | 25.3 ± 1.8 |
| HbE levels (%) | 28.0 ± 2.4 | 28.1 ± 3.2 | 18.7 ± 3.1 | 19.1 ± 3.6 |
| RDW (%) | 13.4 ± 0.8 | 13.4 ± 0.6 | 13.3 ± 0.5 | 13.7 ± 0.3 |

Note: RBC count: red blood cell count, Hb: hemoglobin level, Hct: hematocrit, MCV: mean corpuscular volume, MCH: mean corpuscular hemoglobin, MCHC: mean corpuscular hemoglobin concentration, HbE levels: hemoglobin E levels and RDW: red cell distribution width.

et al. (2005) [19]. In this study, 50 double heterozygotes of HbE and α-thal 1 (SEA) type (26 women and 24 men) were found among 397 individuals receiving care at the Antenatal Care Clinic service at Chiangrai Prachanukroh Hospital between 2017 and 2021. Thus the prevalence of double heterozygotes of HbE and α-thal 1 (SEA) type among HbE carriers was 12.6%. The gene frequency for HbE was 0.470, whereas that for the SEA deletion was 0.059. Being able to distinguish double heterozygotes of HbE and α-thal 1 (SEA) type from pure HbE heterozygote is very important. A woman could conceive a child with Hb Bart's hydrops fetalis but not know until birth, putting the mother at risk of developing complications. However, during the data collection period, no partners at-risk were found (who both carry double heterozygotes of HbE and α-thal 1 (SEA) type or one being double heterozygotes of HbE and α-thal 1 (SEA) type and another carrying heterozygote of α-thal 1 (SEA) type). Nevertheless, if there is a partner at-risk and to allow this partner to independently decide which method to choose to avert and prevent having a child with this severe disease. The goal is to empower this at-risk partner to make a decision on the best method. The key issue is selecting the preferred

**Table 4. Reevaluated using combined cut-offs in samples with Hb < 10.0 g/dL, 10.0- 11.9 g/dL and ≥ 12.0 g/dL (S5 Table).**

| Combined cut-offs | Hb < 10.0 g/dL (n = 10) | Hb 10.0–11.9 g/dL (n = 230) | Hb ≥ 12.0 g/dL (n = 157) |
|---|---|---|---|
| | MCH + HbE level | MCV + MCH + HbE level | MCV + HbE level |
| | 21.0 (pg) + 21.2 (%) | 72.8 (fL) + 23.9 (pg) + 25.6 (%) | 76.7 (fL) + 27.1 (%) |
| Sensitivity (%) | 100 | 100 | 100 |
| Specificity (%) | 100 | 67.7 | 31.2 |
| Positive predictive value (PPV) (%) | 100 | 30.9 | 16.7 |
| Negative predictive value (NPV) (%) | 100 | 100 | 100 |
| Accuracy (%) | 100 | 71.7 | 39.5 |

Note: MCV: mean corpuscular volume, MCH: mean corpuscular hemoglobin and HbE level: hemoglobin E level.

**Table 5. Reevaluated using single cut-offs in samples with Hb < 10.0 g/dL, 10.0-11.9 g/dL and ≥ 12.0 g/dL (S5 Table).**

| Hb < 10.0 g/dL | | | |
|---|---|---|---|
| Single cut-offs | MCH | MCV | HbE level |
| | 21.0 (pg) | 64.9 (fL) | 21.2 (%) |
| Sensitivity (%) | 100 | 100 | 100 |
| Specificity (%) | 100 | 100 | 100 |
| Positive predictive value (PPV) (%) | 100 | 100 | 100 |
| Negative predictive value (NPV) (%) | 100 | 100 | 100 |
| Accuracy (%) | 100 | 100 | 100 |
| **Hb 10.0–11.9 g/dL** | | | |
| Single cut-offs | MCH | MCV | HbE level |
| | 23.9 (pg) | 72.8 (fL) | 25.6 (%) |
| Sensitivity (%) | 75.9 | 86.2 | 93.1 |
| Specificity (%) | 74.1 | 92.0 | 92.0 |
| Positive predictive value (PPV) (%) | 29.7 | 61.0 | 62.8 |
| Negative predictive value (NPV) (%) | 95.5 | 97.9 | 98.3 |
| Accuracy (%) | 77.1 | 91.2 | 92.2 |
| **Hb ≥ 12.0 g/dL** | | | |
| Single cut-offs | MCH | MCV | HbE level |
| | 25.3 (pg) | 76.7 (fL) | 27.1 (%) |
| Sensitivity (%) | 68.4 | 94.7 | 100 |
| Specificity (%) | 33.3 | 63.0 | 61.6 |
| Positive predictive value (PPV) (%) | 12.4 | 26.1 | 26.4 |
| Negative predictive value (NPV) (%) | 88.5 | 98.9 | 100 |
| Accuracy (%) | 37.6 | 66.9 | 66.2 |

Note: MCV: mean corpuscular volume, MCH: mean corpuscular hemoglobin and HbE level: hemoglobin E level.

prenatal diagnosis method and planning for severe thalassemia results, such as terminating the pregnancy. Preliminary assessment using red blood cell indices and HbE levels can help with distinction, aiming for disease prevention or early diagnosis. This study revealed that differences in the MCV, MCH, MCHC and HbE levels between individuals with these

genotypes were statistically significant (mean ± SD): in pure HbE heterozygote, MCV = 78.2 ± 3.9 fL, MCH = 24.6 ± 1.5 pg, MCHC = 27.9 ± 2.8 g/dL and HbE level = 28.1 ± 2.8%, while in double heterozygotes of HbE and α-thal 1 (SEA) type, these values were lower, at MCV = 65.5 ± 7.1 fL, MCH = 22.4 ± 2.9 pg, MCHC = 25.3 ± 2.7 g/dL and HbE level = 18.9 ± 3.4%. The MCV and HbE level were similar to those reported by Sanchaisuriya et al. (2003), who studied 202 Thai subjects diagnosed with HbE heterozygote in combination with various forms of α-thalassemia [30] and reported that the frequency of double heterozygotes of HbE and α-thal 1 (SEA) type was about 10.4%. The hematological parameters measured in this study were as follows (median values): in pure HbE heterozygote, MCV = 78.2 fL (77.5–80.0 fL), MCH = 25.5 pg (25.2–25.8 pg), MCHC = 32.6 g/dL (32.1–32.9 g/dL) and HbE level = 29.2% (28.6–29.4%), while in double heterozygotes of HbE and α-thal 1 (SEA) type, MCV = 69.5 fL (68.4–70.8 fL), MCH = 22.4 pg (21.6–22.9 pg), MCHC = 32.6 g/dL (32.0–33.1 g/dL) and HbE level = 21.4% (20.2–22.3%). Our study is similar to the large cohort study by Charoenkwan et al. (2005), who studied 844 cases HbE carriers among pregnant women and/or their partners who visited the Thalassemia Unit, Department of Pediatrics, Faculty of Medicine, Chiang Mai University, Chiang Mai, Thailand, from 2000 to 2002 [19]. Among 844 cases, 755 were HbE heterozygote and 93 were double heterozygotes of HbE and α-thal 1 (SEA) type; the prevalence of double heterozygotes of HbE and α-thal 1 (SEA) type among HbE carriers was 11%. In her study, the mean ± SD HbE was approximately 23.3 ± 3.1% in HbE heterozygote and 17.0 ± 3.7% in double heterozygotes of HbE and α-thal 1 (SEA) type. Our study is also similar to that of Limveeraprajak E. (2019) [20], who reported a significant decrease in hematological parameters in double heterozygotes of HbE and α-thal 1 (SEA) type (MCV = 68.0 ± 6.8 fL, MCH = 22.0 ± 2.5 pg and HbE level = 20.3 ± 2.9%) compared to those in pure HbE heterozygote (MCV = 77.1 ± 5.8 fL, MCH = 25.4 ± 2.2 pg and HbE level = 26.1 ± 3.5%).

Reevaluation of cut-offs classified according to hemoglobin concentrations was performed. The distinctive point compared to other studies is that this study was conducted on samples that excluded iron-deficiency anemia (due to iron-deficiency anemia, MCV and MCH are lower than normal similar to thalassemia trait and to reduce false positive screening, RDW is good indicator at no additional cost) [24]. There was 100% sensitivity and 100% NPV when using all combined cut-offs (MCH + HbE level is suitable for the group with Hb < 10 g/dL, MCV + MCH + HbE level for the group with Hb = 10.0–11.9 and MCV + HbE level for the group with Hb ≥ 12.0 g/dL. These cut-offs have been evaluated as providing the highest accuracy (Leckngam et al. (2017)) [23]). However, the sensitivity and NPV were reduced when single cut-offs were used in the group with Hb = 10.0–11.9 and Hb ≥ 12.0 g/dL. When using combined cut-offs, the specificity and PPV were 100% and 100% in the group with Hb < 10 g/dL, 67.7% and 30.9% occurred in the group with Hb = 10.0–11.9 g/dL and 31.2% and 16.7% in the group with Hb ≥ 12.0 g/dL, respectively. In contrast, using a single cut-off for MCV and HbE level in the group with Hb ≥ 12.0 g/dL improved the specificity and PPV by 63.0% and 26.1% and 61.6% and 26.4%, respectively. According to the accuracy evaluation, it was found that the combined cut-offs for the group with Hb < 10 g/dL and Hb 10.0–11.9 g/dL demonstrated high accuracy, while the group with Hb ≥ 12.0 g/dL had low accuracy. However, it can be substituted by using a single cut-off of MCV and HbE levels that showed moderate accuracy in samples with Hb ≥ 12.0 g/dL. The practical impact of this study is that it allows the medical laboratory personnel to distinguish pure HbE heterozygote from double heterozygotes of HbE and α-thal 1 (SEA) type more accurately and effectively.

## Conclusions

The prevalence of double heterozygotes of HbE and α-thal 1 (SEA) type among HbE carriers in pregnant women and their partners who visited the Antenatal Care Clinic, Chiangrai Prachanukroh Hospital, from 2017 to 2021 was 12.6%. This value is considered to be quite high. In partners where both are HbE carriers or one carries HbE and another is an α-thalassemia 1 (SEA) type carrier, the correct investigation for coinheritance of SEA type in HbE carriers is especially important for accurate genetic counseling. This study compared hematological parameters between pure HbE heterozygote and double heterozygotes of HbE and α-thal 1 (SEA) type and found that the MCV, MCH, MCHC and HbE level were significantly lower in double heterozygotes of HbE and α-thal 1 (SEA) type, while the RBC and RDW were greater.

Reevaluated cut-offs in samples that excluded iron-deficiency anemia demonstrated that when screening for double heterozygotes of HbE and α-thal 1 (SEA) type, combined cut-offs should be used for samples with Hb < 10.0 g/dL and 10.0–11.9 g/dL, while a single cut-off for HbE level should be used for samples with Hb ≥ 12.0 g/dL.

## Supporting information

**S1 Table. Data collected from pregnant women and their partners (hemoglobin typing EA) 2017-2021.**
(DOCX)

**S2 Methods. Laboratory methods.**
(DOCX)

**S3 Table. Data collected from pregnant women and their partners ( grouping by E and SEA/E).**
(DOCX)

**S4 Table. Data collected from pregnant women and their partners (stratified by gender).**
(DOCX)

**S5 Table. Data collected from pregnant women and their partners (classified by hemoglobin levels).**
(DOCX)

## Acknowledgments

We (Prapapun Leckngam, School of Health Science, Mae Fah Luang University, Chiangrai, Thailand; Phanida Wamontree, School of Integrative Medicine, Mae Fah Luang University, Chiangrai, Thailand and Tapanut Chuaykarn, Chiangrai Prachanukroh Hospital, Chiangrai, Thailand) would like to express gratitude to the director of Chiangrai Prachanukroh Hospital for the opportunity to conduct the research in the hospital. Thanks to the nursing team in the hospital for dedicating their time to collect data. We also thank our advisor for providing consultation and giving valuable information to this research.

## Author contributions

**Conceptualization:** Prapapun Leckngam, Phanida Wamontree, Tapanut Chuaykarn.

**Data curation:** Prapapun Leckngam, Phanida Wamontree, Tapanut Chuaykarn.

**Formal analysis:** Prapapun Leckngam, Phanida Wamontree, Tapanut Chuaykarn.

**Funding acquisition:** Prapapun Leckngam.

**Methodology:** Prapapun Leckngam.

**Project administration:** Prapapun Leckngam.

**Resources:** Tapanut Chuaykarn.

**Validation:** Prapapun Leckngam, Phanida Wamontree, Tapanut Chuaykarn.

**Writing – original draft:** Prapapun Leckngam, Phanida Wamontree, Tapanut Chuaykarn.

**Writing – review & editing:** Prapapun Leckngam.

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
