## [Decision Letter · Decision Letter 0]

16 May 2025

Dear Dr. Leckngam,

Thank you for submitting your manuscript to PLOS ONE. After careful consideration, we feel that it has merit but does not fully meet PLOS ONE’s publication criteria as it currently stands. Therefore, we invite you to submit a revised version of the manuscript that addresses the points raised during the review process.

We look forward to receiving your revised manuscript.

Kind regards,

Mithun Sikdar

Academic Editor

PLOS ONE

**Journal Requirements:**

1. When submitting your revision, we need you to address these additional requirements. Please ensure that your manuscript meets PLOS ONE's style requirements, including those for file naming. The PLOS ONE style templates can be found at https://journals.plos.org/plosone/s/file?id=wjVg/PLOSOne_formatting_sample_main_body.pdf and https://journals.plos.org/plosone/s/file?id=ba62/PLOSOne_formatting_sample_title_authors_affiliations.pdf 2. Thank you for stating the following financial disclosure: Mae Fah Luang UniversityGrant no. 651A05003  Please state what role the funders took in the study.  If the funders had no role, please state: "The funders had no role in study design, data collection and analysis, decision to publish, or preparation of the manuscript." If this statement is not correct you must amend it as needed. Please include this amended Role of Funder statement in your cover letter; we will change the online submission form on your behalf. 3. Thank you for stating the following in the Acknowledgments Section of your manuscript: I would like to express gratitude to the director of Chiangrai Prachanukroh Hospital for the opportunity to conduct this research and to the head of the Antenatal Care Clinic and the nursing team in the hospital for dedicating their time to collect data. I also thank my advisor for consultation and for giving valuable information to this research. Thanks also to Mae Fah Luang University for funding this research. We note that you have provided funding information that is not currently declared in your Funding Statement. However, funding information should not appear in the Acknowledgments section or other areas of your manuscript. We will only publish funding information present in the Funding Statement section of the online submission form. Please remove any funding-related text from the manuscript and let us know how you would like to update your Funding Statement. Currently, your Funding Statement reads as follows: Mae Fah Luang UniversityGrant no. 651A05003  Please include your amended statements within your cover letter; we will change the online submission form on your behalf. 4. One of the noted authors is a group or consortium. In addition to naming the author group, please list the individual authors and affiliations within this group in the acknowledgments section of your manuscript. Please also indicate clearly a lead author for this group along with a contact email address. 5. Your ethics statement should only appear in the Methods section of your manuscript. If your ethics statement is written in any section besides the Methods, please delete it from any other section. 6. Please include captions for your Supporting Information files at the end of your manuscript, and update any in-text citations to match accordingly. Please see our Supporting Information guidelines for more information: http://journals.plos.org/plosone/s/supporting-information.

**Additional Editor Comments:**

I have read the manuscript in detail and appreciate the effort that has gone into the study. I am pleased to note that two reviewers have recommended revision; however, the third reviewer has expressed reservations about publishing the manuscript in PLOS ONE. I trust the authors are willing to revise the manuscript in line with the reviewers’ comments.

The inclusion of couples carrying double heterozygous mutations for HbE and SEA α-thalassemia presents a novel aspect of the study. However, unless the recruitment of couples is clearly justified, the analysis may appear to reflect only individual-level screening. It seems the authors have established substantial rapport with the participating couples, which likely included genetic counselling. If available, pregnancy outcomes from these couples would significantly enrich the dataset and add a novel dimension to the findings.

I encourage the authors to further highlight the novelty of their work. Since contact with participants has already been established, incorporating additional socio-economic variables would enhance the analytical depth of the study.

The manuscript, submitted in February 2025, includes data from individuals recruited between 2017 and 2021. Given this timeline, follow-up records are expected to be available. Since couple-based recruitment is central to the study, I recommend including all available follow-up data to better capture outcomes such as pregnancy loss or neonatal complications—this will considerably strengthen the paper.

Regarding data presentation:

1) Table 1 currently presents data for 397 individuals but lacks couple-based information, which is essential considering the study's focus. Please add a cross-tabulation showing different combinations of hemoglobin variants among couples. Also, the age-related distributions are somewhat inconsistent and should be corrected for clarity.

2) Table 2 compares RBC parameters between single HbE heterozygotes and SEA/E double heterozygotes but does not stratify by sex, which is clinically important.

3) Table 3 should be revised to present sex-specific data.

4) In Lines 259–260, the statement “50 double heterozygotes SEA/E were found among 397 individuals (26 pregnant women and 24 male partners)” is unclear and should be clarified.

5) A map of the study area should be included to provide geographic context.

6) The sentence in Line 292, “validation of cut-offs classified according to hemoglobin concentrations was performed,” should be revised to better reflect the updated analysis and inferences.

7) The supplementary tables should be updated to reflect couple-level (partnered) information rather than individual data.

8) Finally, the title should be reconsidered to better reflect the focus and novelty of the study, in light of the reviewers' comments.

I am appending the reviewers’ comments for your kind attention and necessary action.

**Reviewer 1**

Regarding Question 1. Technical Soundness and Data Support for Conclusions

The manuscript presents a well-conducted study with a sound methodology and appropriate data collection techniques. The data support the conclusions drawn, particularly in terms of the prevalence of SEA/E double heterozygotes and the use of hematological parameters for differentiation. However, the study would benefit from a more explicit discussion of potential confounding variables, such as iron deficiency anemia, which may affect MCV and MCH values.

Recommendation:

Include a discussion on how confounding variables, such as iron deficiency, may have influenced the hematological parameters in the discussion section.

Regarding Question 2. Statistical Analysis

The statistical tests are appropriately applied. However, additional justification for the selected p-value threshold of 0.05 would improve transparency.

Recommendation:

Provide a brief explanation for choosing p<0.05 as the statistical significance threshold in the Methods section, after line 172.

Regarding Question 3. Data Availability

The manuscript states that “all relevant data are within the manuscript and its supporting information files,” but no publicly accessible dataset is provided.

While ethical concerns may restrict sharing, authors should clarify whether anonymized data can be made available upon request.

Recommendation:

Indicate whether anonymized datasets or raw data will be deposited in a public repository, or provide a justification for restrictions.

Regarding Question 4. Manuscript Clarity and English Language Quality

The manuscript is well-structured and easy to follow. However, some grammatical errors and awkward phrasing need revision.

Notable Errors are:

1. Line 23: “If pregnant women who carry this genotype get pregnant with their partners…”

→ Suggested revision: “If pregnant women carrying this genotype conceive with partners…”

2. Line 96: “Due to the poor survival of an affected fetuses…”

→ Suggested revision: “Due to the poor survival rate of affected fetuses…”

3. Line 125: “It’s at this hospital that pregnant women attend very much…”

→ Suggested revision: “This hospital serves a high volume of pregnant women…”

4. Line 280: “Our study is similar to the large cohort study from Charoenkwan P. et al. (2005)…”

→ Suggested revision: “Our study is similar to the large cohort study by Charoenkwan et al. (2005)…”

Recommendation:

Revise these grammatical issues to improve readability.

Additional Recommendations.

1. Quality Control Measures: Explain how quality control was ensured in laboratory procedures for HPLC and PCR testing in the method section, after line 164.

2. Conclusion Expansion: Provide more actionable recommendations on how these findings could influence policy and clinical guidelines in the conclusion section, after line 312.

3. Significance and Validity of Claim: The authors claim that appropriate cut-offs for hematological parameters and HbE levels can distinguish SEA/E carriers from single HbE heterozygotes. While the claims are valid, a stronger comparison with similar studies in other populations would improve the generalizability of the findings.

In summary, the manuscript is well-executed and contributes valuable insights into SEA/E double heterozygotes and prenatal screening strategies. Addressing the suggested revisions will enhance clarity, and compliance with PLOS ONE publication guidelines.

**Reviewer 2**

The manuscript examines the "Prevalence of double heterozygotes SEA/E in pregnancies and their couples who received antenatal care at Chiangrai Prachanukroh Hospital." This research is crucial for identifying pregnant women affected by this genetic condition, as it can play a significant role in allowing healthcare professionals to implement targeted strategies that could help prevent or effectively manage various complications. These complications may extend beyond physical health to include psychological and mental health challenges.

However, the findings and conclusions outlined in the manuscript fall short of establishing clear clinical correlations. There is also a noticeable absence of practical recommendations to guide healthcare providers on how to support these women by utilizing these effective screening measures and management options. It is essential that this vital information be incorporated into both the discussion and the conclusion sections of the manuscript to enhance its practical applicability and relevance for clinical practice.

Title

The study's title, "Prevalence of Double Heterozygotes SEA/E in Pregnancies and Their Couples Who Received Antenatal Care at Chiangrai Prachanukroh Hospital," is confusing. Did the authors intend to use "partners" instead of "couples" in the title? The term "couples" is unclear in this context, and it would be better to clarify what the authors are referring to.

Introduction

Page 4, lines 68-70: The statement seems incomplete with the word "who". Kindly review and make this statement more clearer

Page 5, line 93: There should be a reference to support the figure quoted in the sentence.

Page 5, lines 94-96: There should also be a reference to support the statement made on the risk of perinatal death and associated complications for the mother.

Page 6, lines 123-126: The statement made here should be under the method section. The authors should review this make appropriate correction.

Page 6, lines 128-130: The figure stated here, Figure 1, should be closer to its first mention. Currently, readers need to search for where the figure is referenced in the manuscript.

Methods

Page 6, line 134. The authors should clarify the earlier comments regarding the title, which is also reflected here, particularly concerning "pregnant women and their couples." Do the authors mean "partners" instead? It is important to clarify this for better understanding.

Page 8, lines 178-179: The authors should provide a detailed explanation of the significance of critical parameters, including sensitivity, specificity, positive predictive value (PPV), negative predictive value (NPV), and overall accuracy. Additionally, they need to establish precise grade levels to interpret these metrics accurately. It would be beneficial to specify the thresholds that classify these parameters as low, moderate, and high, enabling readers to understand the implications of varying results more effectively.

Page 10, lines 206-208: The statement made under the validation of cutoffs used should be in the method section. The result section should solely focus on the study's findings.

Page 210-222: It is essential to define the metrics used for validating the cutoffs in the methods section to enhance clarity and help readers understand their significance in the validation process. Additionally, it is important to present the classification levels of these tests—such as low, moderate, or high—so that readers can relate to them more effectively.

Page 11, lines 227-232, For Tables 1 and 2, the authors should define all abbreviations as footnotes under the table.

Page 12 and 13: The authors should define the abbrevaitions on Tables 3 and 4 as well as footnote.

Discussion:

Page 14, lines 264-266: The authors pointed out that mothers affected by this genetic disorder may experience some complications, yet they did not detail what those specific complications might entail. Identifying these complications is essential for proper understanding and management. Furthermore, the authors should elucidate how these complications can be prevented, drawing on the findings of this study. This knowledge is vital for implementing these screening measures that can either avert these complications entirely or ensure they are managed accurately and efficiently when they do occur.

Pages 13-15, lines 254-299: The discussion lacks depth in interpreting this study's findings and seems to merely reiterate the results. Additionally, there is insufficient clinical correlation to these findings. For instance, what are the clinical implications of these results in real-world settings? How might they impact practice, research, and policy? In what ways can they enhance overall clinical practice? Moreover, is there a need for future research in this area? If so, which specific aspects should be explored further? Therefore, the discussion should be refined to integrate these findings better.

Conclusion:

Page 16, lines 310-312: The basis for the recommendation made as well as the recommendation itself did not reflect in the discussion.

Acknowledgment

Page 16, line 315: I believe the manuscript write-up is a joint effort between the authors. Consequently, the opening statement that begins with "I would like to express" fails to capture the essence of this teamwork and the collective dedication to the study. Given the involvement of other authors, it is essential to refrain from using first-person language, as it does not reflect the unified contributions made by the team. Instead, employing third-person language will more accurately highlight the shared insights, perspectives, and expertise of all contributors, underscoring the spirit of collaboration that underpins this work.

**Reviewer 3**

In this paper, describe their study on the hematological and hemoglobin analysis, comparing pure Hb E heterozygote and double heterozygote for Hb E and α-thalassemia 1. Various hematological parameters were described and compared to differentiate the two groups. Study was done retrospectively on selected group of subjects with specified criterions. Unfortunately, the finding is not new to warrant publication. In fact, differentiation of the two groups of Hb E could be made easily using just the level of Hb E (+Hb A2) on Hb analysis. A screening protocol regarding this has been well established and described in the national guideline, being applied throughout the country with high prevalence of thalassemia like Thailand. Other criticisms are listed below.

1. The term “double heterozygotes SEA/E” used throughout the whole manuscript is not suitable especially in the title which should be self-explainable. I believe that readers outside the field of hemoglobinopathy study would not understand this term.

2. As for no. 1, the term “single Hb E heterozygote” is not acceptable. This could mean a single copy of Hb E as seen in heterozygote, but in a double heterozygote SEA/E described in the manuscript, you also have a single copy of Hb E. This could be confusing. I do understand that the authors use the term “single Hb E heterozygote” to describe a pure Hb E heterozygote (i.e., without α-thalassemia). This could be simply read as Hb E heterozygote.

3. Introduction part is too long, containing several unnecessary information,… could be reduced by half. In addition, I do not understand the meaning of the last sentence of the introduction. … “It’s at this hospital that pregnant women attend very much, which accounts for approximately 15% of all pregnancies in Chiangrai province”… Re-writing is needed.

4. It has been known for years that Hb E heterozygote with Hb E < 25% could be co-inherited with α -thalassemia and further DNA analysis of α-thalassemia 1 is recommended. This has been accepted in the general guideline of thalassemia screening in many areas including Thailand. In fact, you have also seen this in your study as shown in the Table 2 of the current paper (28.1+2.8 v.s. 18.9+3.4 and also shown in Figure 2D). This is a very simple differentiation parameter available after Hb analysis. No need to look for any other complicated cut-off values as described in the current paper. The screening protocol regarding this has been published for years in the two papers (Fucharoen G, et al. A simplified screening strategy for thalassaemia and haemoglobin E in rural communities of Southeast Asia. Bulletin World Health Organ 2004; 82: 364-372. and Sanchaisuriya K, et al. A reliable screening protocol for thalassemia and hemoglobinopathies in pregnancy; an alternative approach to electronic blood cell counting. Am J Clin Pathol 2005; 123: 113-118.). Unfortunately, these two papers have not been mentioned in the current paper.

5. Subjects were selectively and retrospectively recruited under specified criterion. They could be regarded as known samples. With this approach, it is not possible to calculate accurate sensitivity, specificity, NPV and PPV. The authors may use the data of these subjects to establish the cut-off value but they need to prove it prospectively on unknown subjects recruited continuously in independent cohorts (at least three is best) before reliable analytical characteristics can be calculated.

6. Table 1 is not necessary. As for Figure 1 which describes well known knowledge of genetic inheritance,… this is no need…. With the above criticisms and minute novel information, I do not think that this paper is suitable for publication in PLoS ONE.

Reviewers' comments:

Reviewer's Responses to Questions

**Comments to the Author**

1. Is the manuscript technically sound, and do the data support the conclusions?

Reviewer #1: Yes

Reviewer #2: Yes

Reviewer #3: No

2. Has the statistical analysis been performed appropriately and rigorously?

Reviewer #1: Yes

Reviewer #2: Yes

Reviewer #3: N/A

3. Have the authors made all data underlying the findings in their manuscript fully available?

Reviewer #1: No

Reviewer #2: No

Reviewer #3: No

4. Is the manuscript presented in an intelligible fashion and written in standard English?

Reviewer #1: Yes

Reviewer #2: Yes

Reviewer #3: No

**Reviewer #1:**  Regarding Question 1. Technical Soundness and Data Support for Conclusions

The manuscript presents a well-conducted study with a sound methodology and appropriate data collection techniques. The data support the conclusions drawn, particularly in terms of the prevalence of SEA/E double heterozygotes and the use of hematological parameters for differentiation. However, the study would benefit from a more explicit discussion of potential confounding variables, such as iron deficiency anemia, which may affect MCV and MCH values.

Recommendation:

Include a discussion on how confounding variables, such as iron deficiency, may have influenced the hematological parameters in the discussion section.

Regarding Question 2. Statistical Analysis

The statistical tests are appropriately applied. However, additional justification for the selected p-value threshold of 0.05 would improve transparency.

Recommendation:

Provide a brief explanation for choosing p<0.05 as the statistical significance threshold in the Methods section, after line 172.

Regarding Question 3. Data Availability

The manuscript states that “all relevant data are within the manuscript and its supporting information files,” but no publicly accessible dataset is provided.

While ethical concerns may restrict sharing, authors should clarify whether anonymized data can be made available upon request.

Recommendation:

Indicate whether anonymized datasets or raw data will be deposited in a public repository, or provide a justification for restrictions.

Regarding Question 4. Manuscript Clarity and English Language Quality

The manuscript is well-structured and easy to follow. However, some grammatical errors and awkward phrasing need revision.

Notable Errors are:

1. Line 23: “If pregnant women who carry this genotype get pregnant with their partners…”

→ Suggested revision: “If pregnant women carrying this genotype conceive with partners…”

2. Line 96: “Due to the poor survival of an affected fetuses…”

→ Suggested revision: “Due to the poor survival rate of affected fetuses…”

3. Line 125: “It’s at this hospital that pregnant women attend very much…”

→ Suggested revision: “This hospital serves a high volume of pregnant women…”

4. Line 280: “Our study is similar to the large cohort study from Charoenkwan P. et al. (2005)…”

→ Suggested revision: “Our study is similar to the large cohort study by Charoenkwan et al. (2005)…”

Recommendation:

Revise these grammatical issues to improve readability.

Additional Recommendations.

1. Quality Control Measures: Explain how quality control was ensured in laboratory procedures for HPLC and PCR testing in the method section, after line 164.

2. Conclusion Expansion: Provide more actionable recommendations on how these findings could influence policy and clinical guidelines in the conclusion section, after line 312.

3. Significance and Validity of Claim: The authors claim that appropriate cut-offs for hematological parameters and HbE levels can distinguish SEA/E carriers from single HbE heterozygotes. While the claims are valid, a stronger comparison with similar studies in other populations would improve the generalizability of the findings.

In summary, the manuscript is well-executed and contributes valuable insights into SEA/E double heterozygotes and prenatal screening strategies. Addressing the suggested revisions will enhance clarity, and compliance with PLOS ONE publication guidelines.

**Reviewer #2:**  The manuscript examines the "Prevalence of double heterozygotes SEA/E in pregnancies and their couples who received antenatal care at Chiangrai Prachanukroh Hospital." This research is crucial for identifying pregnant women affected by this genetic condition, as it can play a significant role in allowing healthcare professionals to implement targeted strategies that could help prevent or effectively manage various complications. These complications may extend beyond physical health to include psychological and mental health challenges.

However, the findings and conclusions outlined in the manuscript fall short of establishing clear clinical correlations. There is also a noticeable absence of practical recommendations to guide healthcare providers on how to support these women by utilizing these effective screening measures and management options. It is essential that this vital information be incorporated into both the discussion and the conclusion sections of the manuscript to enhance its practical applicability and relevance for clinical practice.

Title

The study's title, "Prevalence of Double Heterozygotes SEA/E in Pregnancies and Their Couples Who Received Antenatal Care at Chiangrai Prachanukroh Hospital," is confusing. Did the authors intend to use "partners" instead of "couples" in the title? The term "couples" is unclear in this context, and it would be better to clarify what the authors are referring to.

Introduction

Page 4, lines 68-70: The statement seems incomplete with the word "who". Kindly review and make this statement more clearer

Page 5, line 93: There should be a reference to support the figure quoted in the sentence.

Page 5, lines 94-96: There should also be a reference to support the statement made on the risk of perinatal death and associated complications for the mother.

Page 6, lines 123-126: The statement made here should be under the method section. The authors should review this make appropriate correction.

Page 6, lines 128-130: The figure stated here, Figure 1, should be closer to its first mention. Currently, readers need to search for where the figure is referenced in the manuscript.

Methods

Page 6, line 134. The authors should clarify the earlier comments regarding the title, which is also reflected here, particularly concerning "pregnant women and their couples." Do the authors mean "partners" instead? It is important to clarify this for better understanding.

Page 8, lines 178-179: The authors should provide a detailed explanation of the significance of critical parameters, including sensitivity, specificity, positive predictive value (PPV), negative predictive value (NPV), and overall accuracy. Additionally, they need to establish precise grade levels to interpret these metrics accurately. It would be beneficial to specify the thresholds that classify these parameters as low, moderate, and high, enabling readers to understand the implications of varying results more effectively.

Page 10, lines 206-208: The statement made under the validation of cutoffs used should be in the method section. The result section should solely focus on the study's findings.

Page 210-222: It is essential to define the metrics used for validating the cutoffs in the methods section to enhance clarity and help readers understand their significance in the validation process. Additionally, it is important to present the classification levels of these tests—such as low, moderate, or high—so that readers can relate to them more effectively.

Page 11, lines 227-232, For Tables 1 and 2, the authors should define all abbreviations as footnotes under the table.

Page 12 and 13: The authors should define the abbrevaitions on Tables 3 and 4 as well as footnote.

Discussion:

Page 14, lines 264-266: The authors pointed out that mothers affected by this genetic disorder may experience some complications, yet they did not detail what those specific complications might entail. Identifying these complications is essential for proper understanding and management. Furthermore, the authors should elucidate how these complications can be prevented, drawing on the findings of this study. This knowledge is vital for implementing these screening measures that can either avert these complications entirely or ensure they are managed accurately and efficiently when they do occur.

Pages 13-15, lines 254-299: The discussion lacks depth in interpreting this study's findings and seems to merely reiterate the results. Additionally, there is insufficient clinical correlation to these findings. For instance, what are the clinical implications of these results in real-world settings? How might they impact practice, research, and policy? In what ways can they enhance overall clinical practice? Moreover, is there a need for future research in this area? If so, which specific aspects should be explored further? Therefore, the discussion should be refined to integrate these findings better.

Conclusion:

Page 16, lines 310-312: The basis for the recommendation made as well as the recommendation itself did not reflect in the discussion.

Acknowledgment

Page 16, line 315: I believe the manuscript write-up is a joint effort between the authors. Consequently, the opening statement that begins with "I would like to express" fails to capture the essence of this teamwork and the collective dedication to the study. Given the involvement of other authors, it is essential to refrain from using first-person language, as it does not reflect the unified contributions made by the team. Instead, employing third-person language will more accurately highlight the shared insights, perspectives, and expertise of all contributors, underscoring the spirit of collaboration that underpins this work.

**Reviewer #3:**  In this paper, Leckngam P, et al describe their study on the hematological and hemoglobin analysis, comparing pure Hb E heterozygote and double heterozygote for Hb E and α-thalassemia 1. Various hematological parameters were described and compared to differentiate the two groups. Study was done retrospectively on selected group of subjects with specified criterions. Unfortunately, the finding is not new to warrant publication. In fact, differentiation of the two groups of Hb E could be made easily using just the level of Hb E (+Hb A2) on Hb analysis. A screening protocol regarding this has been well established and described in the national guideline, being applied throughout the country with high prevalence of thalassemia like Thailand. Other criticisms are listed below.

1. The term “double heterozygotes SEA/E” used throughout the whole manuscript is not suitable especially in the title which should be self-explainable. I believe that readers outside the field of hemoglobinopathy study would not understand this term.

2. As for no. 1, the term “single Hb E heterozygote” is not acceptable. This could mean a single copy of Hb E as seen in heterozygote, but in a double heterozygote SEA/E described in the manuscript, you also have a single copy of Hb E. This could be confusing. I do understand that the authors use the term “single Hb E heterozygote” to describe a pure Hb E heterozygote (i.e., without α-thalassemia). This could be simply read as Hb E heterozygote.

3. Introduction part is too long, containing several unnecessary information,… could be reduced by half. In addition, I do not understand the meaning of the last sentence of the introduction. … “It’s at this hospital that pregnant women attend very much, which accounts for approximately 15% of all pregnancies in Chiangrai province”… Re-writing is needed.

4. It has been known for years that Hb E heterozygote with Hb E < 25% could be co-inherited with α -thalassemia and further DNA analysis of α-thalassemia 1 is recommended. This has been accepted in the general guideline of thalassemia screening in many areas including Thailand. In fact, you have also seen this in your study as shown in the Table 2 of the current paper (28.1+2.8 v.s. 18.9+3.4 and also shown in Figure 2D). This is a very simple differentiation parameter available after Hb analysis. No need to look for any other complicated cut-off values as described in the current paper. The screening protocol regarding this has been published for years in the two papers (Fucharoen G, et al. A simplified screening strategy for thalassaemia and haemoglobin E in rural communities of Southeast Asia. Bulletin World Health Organ 2004; 82: 364-372. and Sanchaisuriya K, et al. A reliable screening protocol for thalassemia and hemoglobinopathies in pregnancy; an alternative approach to electronic blood cell counting. Am J Clin Pathol 2005; 123: 113-118.). Unfortunately, these two papers have not been mentioned in the current paper by Leckngam P, et al.

5. Subjects were selectively and retrospectively recruited under specified criterion. They could be regarded as known samples. With this approach, it is not possible to calculate accurate sensitivity, specificity, NPV and PPV. The authors may use the data of these subjects to establish the cut-off value but they need to prove it prospectively on unknown subjects recruited continuously in independent cohorts (at least three is best) before reliable analytical characteristics can be calculated.

6. Table 1 is not necessary. As for Figure 1 which describes well known knowledge of genetic inheritance,… this is no need….

**Do you want your identity to be public for this peer review?** For information about this choice, including consent withdrawal, please see our Privacy Policy

Reviewer #1: **Yes: ** Chibuzor Stella Amadi

Reviewer #2: No

Reviewer #3: No

---

## [Decision Letter · Decision Letter 1]

5 Aug 2025

Dear Dr. Leckngam,

Thank you for submitting your manuscript to PLOS ONE. After careful consideration, we feel that it has merit but does not fully meet PLOS ONE’s publication criteria as it currently stands. Therefore, we invite you to submit a revised version of the manuscript that addresses the points raised during the review process.

We look forward to receiving your revised manuscript.

Kind regards,

Mithun Sikdar

Academic Editor

PLOS ONE

Journal Requirements:

Additional Editor Comments:

As the original reviewers were unavailable to confirm their acceptance, a second round of peer review was conducted, which yielded a positive outcome. I kindly request you to revise your manuscript for a final evaluation, after which it may be considered for publication. Additionally, I urge you to carefully revise the English language with the help of professional editorial support.

Reviewers' comments:

Reviewer's Responses to Questions

**Comments to the Author**

Reviewer #4: (No Response)

Reviewer #5: (No Response)

2. Is the manuscript technically sound, and do the data support the conclusions?

Reviewer #4: Yes

Reviewer #5: Partly

3. Has the statistical analysis been performed appropriately and rigorously?

Reviewer #4: Yes

Reviewer #5: Yes

4. Have the authors made all data underlying the findings in their manuscript fully available?

Reviewer #4: Yes

Reviewer #5: Yes

5. Is the manuscript presented in an intelligible fashion and written in standard English?

Reviewer #4: Yes

Reviewer #5: Yes

Reviewer #4: The manuscript offers a compelling and methodologically sound contribution to the field of prenatal genetic screening, particularly in the context of thalassemia control in northern Thailand. The study addresses an important public health issue by evaluating the diagnostic accuracy of hematological parameters in distinguishing HbE heterozygotes from compound heterozygotes of HbE and α-thalassemia 1 (SEA type), which has significant implications for the prevention of severe conditions such as Hb Bart’s hydrops fetalis. A major strength of the study is the careful exclusion of subjects with elevated red cell distribution width (RDW), effectively controlling for the confounding effect of iron deficiency anemia. This step strengthens the internal validity of the findings and reflects careful attention to potential biases.

The use of adjusted hematological cut-off values is particularly noteworthy. These refined parameters improve diagnostic specificity and sensitivity, enhancing the potential utility of routine blood indices as a frontline screening method in resource-limited settings. The authors’ conclusions are well-supported by the data and align with broader goals in reproductive health and genetic counseling in high-prevalence regions.

Areas for improvement

1. Language and Grammar: Some sections contain verbose or awkward sentence constructions. A professional language edit is recommended to improve fluency and precision.Sentences are too long or wordy, often using more words than necessary. This can make the meaning unclear or tiring for the reader. Example (Verbose): “The data which was collected over a period of ten years was thoroughly analyzed using a number of statistical methods which included…”

Improved: “Data collected over ten years were analyzed using multiple statistical methods…”

Phrases or sentences that may sound unnatural, unclear, or grammatically incorrect, even if the meaning is somewhat understandable.

Example: “The outcomes are according to what was expected due to the previous findings.”

Improved: “The outcomes align with previous findings.”

2.Several abbreviations (e.g., MCV, MCH, RDW, Hb Bart’s) are used extensively. Ensure each is clearly defined on first use and consistently applied thereafter to avoid redundancy and confusion.

Figures and Supplementary Material: All figures and supplementary tables must be referenced accurately within the main text. Their accessibility and interpretability should be ensured, especially if they form part of the data validation.

3.Clinical Context: Consider briefly expanding on the practical implications of these findings for frontline health workers or local healthcare systems. This would contextualize the results for implementation.

Overall, this manuscript is of high quality and is well-aligned with the goals of the journal. After minor revisions related to language, formatting, and clarification, it should be suitable for publication.

Reviewer #5: The paper is well framed highlighting the prevalence of double heterozygotes of HbE and α-thal 1 (SEA) among antenatal women in Thailand which plays a key role in identifying couples with this genotype and offering genetic counselling to reduce the risk of bearing an affected child. The authors made an effort to stratify the hematological parameters to differentiate between the pure HbE heterozygotes and double heterozygotes of HbE and α-thal 1 (SEA) which is cost and time effective. Inclusion of RDW to rule out iron deficiency anemia is a good indicator. However, the authors should have evaluated this including serum ferritin which they considered as their limitation. The authors excluded iron deficiency anemia in their samples which adds novelty to the paper.

Recommendations and Revisions suggested:

1. Has the author sought informed consent from the 397 individuals enrolled in the study for screening? If so, include the statement of informed consent from the antenatal couples enrolled in the study under methodology section.

2. Include the statistics used to determine the cut offs for MCV, MCH and HbE levels in the methodology section.

3. In section Discussion, the authors have determined the single and combined cut offs in three different groups based on Hb levels. But it would be better if they explain why MCH + HBE cut offs were considered for Hb<10g/dl grp, MCV+MCH+HbE were considered for Hb-10-11.9 and MCV+HbE for Hb>12g/dl. Also specify how these cut offs can be considered as significant markers in differentiating pure HbE heterozygotes from double heterozygotes of HbE and α-thal 1 (SEA).

Minor Revisions: As per Revised Manuscript

In Line 38, replace expand the term double SAE/E

In Line 39, replace the term reevaluated with reevaluation

In Line 82, replace the word ethic to ethnic

In Line 82, sentence” in a studied by Tatu T et al” should be corrected as “In a study by…….”

In Line 131, reframe the sentence “This retrospective study …………records.

In Line 179 and 257 replace the term Reevaluated to Reevaluation

In Line 271, check the statement Similar to the results obtained when using MCV. Clarify whether it is MCV or MCH.

The paper can be considered for publication following incorporation of the recommendations and revisions suggested above.

**Do you want your identity to be public for this peer review?** For information about this choice, including consent withdrawal, please see our Privacy Policy

Reviewer #4: **Yes: ** Chinwendu Ubani

Reviewer #5: No

---

## [Author Response · Author response to Decision Letter 2]

27 Aug 2025

We have attached a file of the response to reviewers.

---

## [Decision Letter · Decision Letter 2]

19 Sep 2025

Prevalence of double heterozygotes of HbE and α-thal 1 (SEA) type in pregnancies and their partners that received Antenatal Care at Chiangrai Prachanukroh Hospital and reevaluated the cut-offs for differentiation

PONE-D-25-05285R2

Dear Dr. Leckngam,

We’re pleased to inform you that your manuscript has been judged scientifically suitable for publication and will be formally accepted for publication once it meets all outstanding technical requirements.

Kind regards,

Mithun Sikdar

Academic Editor

PLOS ONE

Additional Editor Comments (optional):

Reviewer #5:

Reviewers' comments:

Reviewer's Responses to Questions

**Comments to the Author**

Reviewer #5: All comments have been addressed

2. Is the manuscript technically sound, and do the data support the conclusions?

Reviewer #5: Yes

3. Has the statistical analysis been performed appropriately and rigorously?

Reviewer #5: Yes

4. Have the authors made all data underlying the findings in their manuscript fully available?

Reviewer #5: Yes

5. Is the manuscript presented in an intelligible fashion and written in standard English?

Reviewer #5: Yes

Reviewer #5: As the authors of the manuscript entitled “Prevalence of double heterozygotes of HbE and α-thal 1 (SEA) type in pregnancies and their partners that received Antenatal Care at Chiangrai Prachanukroh Hospital and reevaluated the cut-offs for differentiation” have carefully addressed and responded to all the comments, I recommend publication of the manuscript in PLOS one.

**Do you want your identity to be public for this peer review?** For information about this choice, including consent withdrawal, please see our Privacy Policy

Reviewer #5: No

---

## [Editor Report · Acceptance letter]

PONE-D-25-05285R2

PLOS ONE

Dear Dr. Leckngam,

I'm pleased to inform you that your manuscript has been deemed suitable for publication in PLOS ONE. Congratulations! Your manuscript is now being handed over to our production team.

Kind regards,

on behalf of

Dr. Mithun Sikdar

Academic Editor

PLOS ONE